# Linear Mixed-Effects Model to Quantify the Association between Somatic Cell Count and Milk Production in Italian Dairy Herds

**DOI:** 10.3390/ani13010080

**Published:** 2022-12-25

**Authors:** Tiantian Luo, Wilma Steeneveld, Mirjam Nielen, Lucio Zanini, Alfonso Zecconi

**Affiliations:** 1Faculty of Veterinary Medicine, Department of Population Health Sciences, Section of Farm Animal Health, Utrecht University, 3584 CA Utrecht, The Netherlands; 2Associazione Regionale Allevatori Lombardia, 26013 Crema, Italy; 3One Health Unit, Department of Biomedical, Surgical and Dental Sciences, University of Milan, 20133 Milan, Italy

**Keywords:** somatic cell count, milk yield, mastitis, costs

## Abstract

**Simple Summary:**

Milk production loss due to mastitis in dairy herds is economically important. This study based on milk test records on more than 800,000 cows in Lombardy (Italy), and using a mixed-effects model with six fixed effects (geographical Area, Breed, Days in Milk, Parity, Season and Year) this study confirmed a negative association of somatic cell counts with milk production. However, the changes in milk production were different from others reported from previous scientific literature, suggesting that local factors may affect this association. Therefore, before estimating the economic impacts of mastitis, it is crucial to quantify the association between mastitis and milk production in relation to the characteristics of the sampled population. The results could help in prioritizing the interventions from the advisory services. They may be also used as a reference for areas in other countries with similar characteristics to Lombardy.

**Abstract:**

Milk production loss due to mastitis in dairy herds is economically important. Before estimating the economic impacts of mastitis, it is crucial to quantify the association between mastitis and milk production. The objective of this study was to estimate the association between somatic cell count (SCC, as an indicator of intramammary infection due to mastitis) and milk production for dairy cows in Lombardy, Italy. The test-day (TD) records data of 3816 dairy herds located in three different geographical areas of Lombardy from January 2016 to December 2018 were used. After data editing, the final dataset comprised 10,445,464 TD records from 2970 farms and 826,831 cows. The analysis was carried out by using a mixed-effects model with six fixed effects (geographical Area, Breed, Days in Milk, Parity, Season and Year) and nested random effects for each cow and herd. The results confirmed that the SCC had a negative association with milk production. On average, this study found that any two-fold increase of SCC resulted in a milk production loss of 0.830 (95% CI: −0.832, −0.828) kg/cow/day in the whole of Lombardy. These results can be used for economic calculations on the costs of mastitis.

## 1. Introduction

Bovine mastitis is one of the most frequently occurring and costly diseases affecting the welfare of dairy cows [1,2,3]. The total costs of mastitis include production losses, prevention and treatment costs, culling, changes in product quality and the risk of other diseases [4]. It is known that these costs can vary considerably between farms [5,6,7]. Van Soest et al. [7] reported that the average total cost of mastitis can reach EUR 240/lactating cow per year. In Italy, the average cost of a single clinical mastitis case was estimated at EUR 177 [8], and a decrease of average milk yield per cow of around 2 kg/d was estimated in herds with contagious pathogens when compared with contagious-pathogen-free herds [9]. 

To estimate the economic impacts of mastitis, it is crucial to quantify the production losses for both clinical and subclinical mastitis (SCM), since production losses are considered one of the greatest contributors to the total costs of mastitis [7]. The production losses due to SCM can be determined with the somatic cell count (SCC) [10]. SCC is one of the standard diagnostic tests to detect SCM, and at most farms is measured on a monthly basis at cow level as part of the milk recording system. A value higher than 200,000 cells/mL is considered positive for SCM [11,12]. An increased SCC is thus an indication of an inflammatory reaction, which can substantially reduce milk production. Milk production loss associated with increasing SCC has been investigated by several studies [13,14,15,16]. Halasa et al. [13] reported that when the SCC increased 2-fold after a low SCC period, the milk production decreased by 0.38 and 0.46 kg/d for primiparous and multiparous cows, respectively. Hand et al. [14] found that the daily milk loss ranged from 0.35 to 4.70 kg for SCC values from 200,000 cells/mL to 2,000,000 cells/mL, and the whole lactation milk loss ranged from 165 to 919 kg per lactation. In Brazil, Goncalves et al. [15] revealed that milk losses per unit increase of log-transformed SCC varied between 0.55 kg/d and 2.45 kg/d in three different lactation stages and three parities. The association between SCC and milk production was thus estimated, but mostly on relatively large-scale intensive dairy herds, and specific management in North-West European and American circumstances [15,16,17]. This association is much less determined for small-scale dairy herds or in areas where dairy herds are less intensive (e.g., mountainous areas). Furthermore, it has not been quantified in the Lombardy region of Italy. Lombardy is the leading milk-producing region in Italy, with more than 40% of the national production, and the dairy herds in Lombardy are in different geographical areas (Alps, sub-Alps and Po Valley) with both large-scale intensive dairy herds and small-scale dairy herds (see Appendix A for a regional map describing the different areas). 

The objective of this study was to estimate the association between SCC and milk production for dairy cows in Lombardy, due to the importance of milk production in this area and the absence of specific recent studies on the association. Our hypothesis is that the association may have a different magnitude when compared to other studies, due to the different and peculiar characteristics of dairy herds in the different geographical areas of Lombardy. If this hypothesis were confirmed, the results could help in prioritizing interventions [9] from the advisory services (i.e., from regional breeder associations). They may be also used as a reference for areas in other countries with similar characteristics to Lombardy (i.e., other European countries, South America). 

## 2. Materials and Methods

### 2.1. Study Data

These data were collected from nine provinces including all the three main geographical areas of the region: Alps, sub-Alps and Po Valley. The initial data included 3816 herds and 905,119 cows with 11,396,685 test-day (TD) records from January 2016 to December 2018. 

### 2.2. Milk Sampling and Analysis

The data to be analyzed includes all herds in Lombardy associated with Italian Breeder Association (AIA) and applying routine milk record sampling for three years. Individual cow samplings were performed by certified methods currently applied by the AIA at the laboratories of Regional Breeders Association of Lombardy (ARAL). Samples were taken about every 5 weeks, delivered refrigerated to ARAL labs the same day, and analyzed within 30 h of sampling. SCC analysis was performed by certified methods, currently applied by AIA at the laboratories of ARAL on Fossomatic FC (Foss DK). Cow and milk test records (MTRs) were supplied by AIA through ARAL, and they were: herdID, cowID, number of lactations, SCC and milk yield at every milk test conducted.

### 2.3. Data Editing

The data editing procedure is presented in Figure 1. First, TD records with missing values on SCC were excluded. Secondly, herds with size less than 30 cows were excluded in sub-Alps and Po Valley. In the Alps area, herds with less than 10 cows were excluded. These herds were excluded to avoid a bias related to poor performance and irregular MTR being marginals herds in the different areas, and to make the model results more accurate for the commercial dairy herds. Thirdly, the TD records with SCC higher than 9,999,000 cells/mL were excluded. Fourthly, TD records with Days in Milk (DIM) longer than 400 days were excluded. Finally, if the cow only had one test in a lactation, these records were also excluded. The final dataset comprised 10,445,464 TD records from 2970 farms and 826,831 cows. 

The geographical areas of the herd location were categorized into three groups: Alps, sub-Alps and Po Valley. There were 24 breeds, including Holstein, Brown Swiss, Simmental, Jersey, local breeds and mixed breeds. Because some of the breeds have a higher capability of milk production and tend to have higher SCC [10], the breeds were categorized into three groups: Holstein, Brown Swiss and other breeds. The distribution among the three geographical areas is reported in Table 1. The DIM were categorized into 13 stages, with 30-day intervals. The parity was categorized into three groups: parity 1, parity 2 and parity equal to or greater than 3. According to the date of each TD record, four season groups (April, May and June as spring; July, August and September as summer; October, November and December as autumn; January, February and March as winter) and three year groups (2016, 2017 and 2018) were categorized.

### 2.4. Statistical Analysis

Given the multilevel structure of the longitudinal data, a linear mixed-effects model was used to estimate the association between SCC and milk production. SCC is a right-skewed variable, and this violated the assumption of the linear model. Therefore, the somatic cell count (SCC) was converted to linear somatic cell score using base 2 log transformation of SCC (SCS), and used to fit in the linear model [18]. Several explanatory variables might also influence milk production, and were inserted in the model as fixed effects. These fixed effects included Area (categorical, 3 levels), Breed (categorical, 3 levels), DIM (categorical, 13 levels), Parity (categorical, 3 levels), Season (categorical, 4 levels) and Year (categorical, 3 levels). The TD records were collected over the course of 3 years, with multiple observations per cow and farm. For individual cows, the daily milk production values were correlated. Furthermore, the cows within the same herd were correlated. Therefore, the nested random effects (random intercepts) for each cow and herd were introduced into the model. The model applied is as follows: $$y_{hjklmnop}=\beta_{0}+\beta_{1}\times Log_{2}\left(SCC\right)+\beta_{2}\times Area_{h}+\beta_{3}\times Breed_{j}+\beta_{4}\times DIM_{k}+\beta_{5}\times Parity_{l}+\beta_{6}\times Season_{m}+\beta_{7}\times Year_{n}+Cow_{o}\left(Herd_{p}\right)+\varepsilon_{hjklmnop}$$
where *Y*_hjklmnop_ was the milk production for each cow o in herd p in area h, breed j, DIM class k, number of parity l, season m and year n; β_0_ was the overall mean of milk production; β_1_ was the regression coefficient of the binary logarithm of the SCC × 10^3^ cells/mL; β_2_ was the regression coefficient of the h^th^ class of Area; β_3_ was the regression coefficient of the j^th^ class of Breed; β4 was the regression coefficient of the k^th^ class of DIM; β_5_ was the regression coefficient of the l^th^ class of Parity; β_6_ was the regression coefficient of the m^th^ class of Season; β_7_ was the regression coefficient of the n^th^ class of Year; Cow o and Herd p were the random effects; ε_hjklmnop_ was the residual error. Log2 (SCC), Area, Breed, DIM, Parity, Season and Year were the fixed effects of milk production.

Model selection was based on the Akaike information criterion (AIC). Moreover, the model assumptions were checked for multicollinearity with Variance Inflation Factors (VIFs), normality of residuals, homoscedasticity, homogeneity of variance and influential observations [19]. For the performance of the mixed models, the marginal R-squared measured the proportion of the variance that was explained by the fixed effects. The conditional R-squared, in contrast, measured the proportion of the total variance that was explained by both fixed and random effects in the full model. Similar to R-squared, the Intraclass Correlation Coefficients (ICCs) also provided information on the explained variance, which was explained by the grouping structure in the population. These indexes were all reported after fitting the models.

All computations were performed with R (R Core Team, 2020). The restricted maximum likelihood (REML) estimates of the parameters in linear mixed-effects models were determined using the lmer function in the lme4 [20] package. The fit of the model was tested by the performance (0.7.0) package. Dplyr (0.8.5), lubridate (1.7.9) and ggplot2 (3.3.0) were used for data editing and data visualization.

## 3. Results

### 3.1. Descriptive Analysis

Descriptive statistics of the final dataset are presented in Table 2. Among Lombardian dairy herds, the median herd size was 176 cows, with an average herd size of 228 cows, and a range from 10 to 1099 cows. The average herd sizes in the Alps, sub-Alps and Po Valley were 86, 109 and 233 cows, respectively. The average milk production level at the TD records level was 32.8 kg/d and had a range from 0.2 to 109.7 kg/d. The average SCC at the TD records level was 308 × 10^3^ cells/mL, ranging from 1 to 9999 × 10^3^ cells/mL. 

Milk production for different breed groups of cows was presented in Figure 2. The average milk production of Holstein cows was 33.1 kg/d, which was the highest among all breed groups. Brown Swiss cows produced 25.6 kg/d milk on average, and other breeds of cows produced 29.3 kg/d milk on average. 

SCS for different milk production levels and breed groups are presented in Figure 3 and Figure 4. SCS decreased when milk production increased. Holstein cows had the lowest SCS among all breeds. 

In different areas and cow breeds, the average milk production decreased as SCC increased (see Figure 5 and Figure 6). In the Alps, however, the average milk production decreased more with higher SCC than in other areas. When SCC ≥ 400,000 cells/mL, this decreasing trend of milk production was no longer obvious. 

### 3.2. Statistical Analysis 

The fixed effects estimate of the final linear mixed effect model on the association between SCC and milk production are presented in Table 3. The overall mean of milk production was 34.75 (95% confidence interval (CI): 34.59, 34.91) kg/cow/day at the reference level (in the Po Valley area, Holstein as the breed, the first month of lactation, the first parity, autumn and 2016). For every unity of Log2 SCC increase, the milk production decreased by 0.83 (95% CI: −0.832, −0.828) kg/cow/day. In the Alps area, the milk production was the lowest. Holstein cows had the highest milk production among breeds, while Brown Swiss cows had the lowest milk production. Cows produced more milk in the second or higher parities than in their first parity. Autumn was the lowest milk production season. Milk production increased gradually within the period of the study (2016–2018).

### 3.3. The Fit of the Model 

The VIFs of the fixed independent variables were all low (lower than 1.25), indicating that the correlation between fixed independent variables was low. The marginal R-squared and the conditional R-squared were 0.31 and 0.67, respectively. The residual variability was explained by the random intercept in the model, with an ICC of 0.52. There is no evidence indicating the presence of influential observations. From the visual inspection of the studentized residuals normal Q–Q plot, the normality of the residuals from the model was acceptable. 

When the milk production was greater than 35 kg/day, there was an increasing non-linear relationship between predictor variables and the outcome variable. However, when the milk production was less than 35 kg/day, the model was able to capture the linear relationship (see Appendix A). In Appendix A, the square root of standard residuals and fitted values are plotted to check the assumption of homoscedasticity. There was an increasing level of heteroscedasticity when the fitted value was greater than 40 kg/day of milk.

## 4. Discussion

In this study, the association between SCC and milk production was quantified for Lombardy dairy farms from 2016 to 2018. Results indicated that for every two-fold increase in SCC, there was a loss in milk production of 0.830 kg/cow/day in the whole of Lombardy. The current study also revealed clear differences among the areas in Lombardy. In the Alps, the average herd size was the smallest (86 cows) with the highest average SCC (319,000 cells/mL) and the lowest average milk production (26.3 kg/d). In the Po Valley, the average herd size was the largest (233 cows) with an average SCC of 308,000 cells/mL and the highest milk production (32.9 kg/d). The average herd size in the sub-Alps was 109 cows, with an average SCC of 274,000 cells/mL and an average milk production of 31.5 kg/d. The results of the final model confirmed these differences between regions. Compared with the Po Valley (as the reference in the model), the coefficient estimates (β) of Alps and sub-Alps were −1.354 kg/d and −6.750 kg/d, respectively. These differences may be due to the different herd characteristics between Po Valley, sub-Alps and Alps dairy herds [21]. Despite the differences in the amount of milk production and the SCC levels among areas and breeds, the presence of a strong association between these two parameters was found. Hence, Figure 5 and Figure 6 showed that when SCC increased from 50,000 cells/mL to 400,000 cells/mL, milk production decreased by about 5 kg/d, independently of area and breed. 

For every two-fold increase in SCC, the estimated decrease in milk production was higher in Lombardy than in other European countries. There, the decrease in milk production ranged from 0.2 kg/d to 0.5 kg/d with different mastitis conditions in the Netherlands, Sweden and other countries [13,17,22]. Moreover, in the current study, crude SCC was used to fit into the models, and was not adjusted with the dilution effect [23], so the adjusted SCC could be higher than the observed values. 

Other factors might have influenced the results of the current study. Firstly, the average milk production and SCC in Lombardy were relatively high compared to other studies from different countries. For instance, the average milk production ranged from 23.2 kg/d to 28.3 kg/d, and the average SCC ranged from 65,000 cells/mL to 105,000 cells/mL in the Netherlands [13,24]. The high milk production in Lombardy was due to several reasons. Lombardy dairy herds were genetically specialized with high-milk-production cows [25]. The farmers applied complex diets [26] by using the total mixed ration method perennially, without pasturing. Moreover, the farmers were highly motivated to achieve higher production because of the thin difference between feeding cost and revenues, thus requiring a higher efficiency in order to make a profit [27]. 

Secondly, Lombardy herds have different breeds, while other studies mainly analyzed herds with only Holstein Friesian cows [18,28]. Cows in different breed groups had heterogeneous milk production and SCC levels. Holstein cows had the highest milk production and the lowest SCC among all breed groups, while Brown Swiss cows had the lowest milk production and the highest SCC among all breed groups (Figure 6). It is worth noting that in different areas, the components of the categorical factor “other breeds” group were different. In the Alps area, a substantial number of the local breeds and Simmental were in this group, while in other Lombardy areas, the mixed breed with Holstein was the main component in the “other breeds” group. Since the milk production capability of the local breeds and Simmental was lower than the mixed breed with Holstein cows, in the Alps area, the cows in the “other breeds” group had the lowest milk production and the highest SCC compared with Holstein cows and Brown Swiss cows. 

Finally, autumn was the lowest milk production season in Lombardy (including the Alps area), which was due to the ambient temperature and the calving pattern. In Lombardy, all herds applied similar calving pattern, with cows starting to be dried off in the autumn, which caused the lowest milk production. This finding was different from other studies [29,30] where summer or spring was the lowest production season. 

The diagnosis plots of the model indicate that heteroscedasticity was present when the fitted values were higher than 40 kg/d. This means higher fitted values had larger residuals, and the model did not fit well with larger predicted means. Klein et al. (2016) revealed that the presence of heteroscedasticity could be due to missing some confounders or interaction terms in the model. However, in the current study, these potential confounders that may influence the milk production (e.g., the feeds and farming techniques) were controlled by the random Herd effect in the models. Moreover, the different interaction terms, different random slopes and single parity data were checked by inserting them into the model; however, the heteroscedasticity was not solved. For simplifying the model and for the purposes of this study, these interaction terms and random slopes were not included in the final model.

Two potential consequences could be present due to heteroscedasticity: the least-squares estimator is still a linear and unbiased estimator, but with wider variance; and the standard errors computed for the least-squares estimators are incorrect [31]. This can affect confidence intervals and hypothesis testing that use those standard errors, which could result in misleading conclusions. However, due to the massive data size in the current study, the confidence intervals were all relevantly very small, and this misclassification error should not happen. Furthermore, Schielzeth et al. [32] proved the robustness of linear mixed-effects models. The results showed that the fixed effect estimates in particular were relatively unbiased when heteroscedasticity data were fitted in linear mixed-effects models. Thus, the association between SCC and milk production in Lombardy and the Alps area should be unbiased by using the mixed-effects model.

Some biases might be present in this study. The database was provided by the dairy farmer association in Lombardy. This might lead to selection bias since there is lacking randomness in the samples to represent all the dairy cows in Lombardy or Northern Italy. Moreover, this study excluded the small herds, which may have led to additional selection bias. This selection bias might have caused an underestimation of the loss of milk production, since these small herds had higher SCC and lower milk production. 

Furthermore, the assessment of this association in the Lombardy herds suggested that applying values from studies performed in other countries may be misleading, since the drop in yield as SCC increased was higher. This information should be considered when the economic impact of subclinical mastitis (defined by SCC) is estimated to identify priorities in the application of herd management programs.

## 5. Conclusions

This study focused on quantifying the association between SCC and milk production, by analyzing the TD testing records data of 2970 Lombardy dairy herds from January 2016 to December 2018. The results confirmed that the SCC had a negative association with milk production. On average, this study found that any two-fold increase of SCC resulted in a milk production loss of 0.830 (95% CI: −0.832, −0.828) kg/cow/day in the whole of Lombardy. The pattern of the relationship between SCC and milk production showed that when SCC increased from 50,000 cells/mL to 400,000 cells/mL, milk production decreased by about 5 kg/d, independently of area or breed. However, outside this range, significant differences may be observed. The results also confirmed that an improvement in herd udder health would result in a significant increase in milk yield and, therefore, of herd efficiency. The magnitude of this increase was shown to be different from those provided by previous studies, suggesting the importance of assessing these aspects specifically in the different production areas.

## Figures and Tables

**Figure 1 animals-13-00080-f001:**
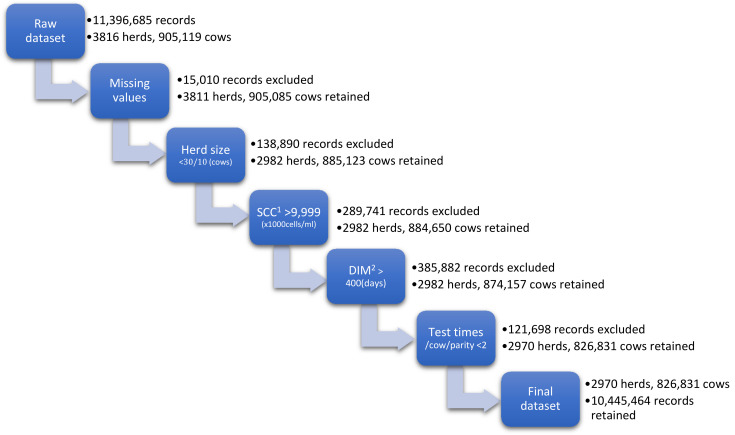
Editing criteria and number of herds, cows and test day records retained and excluded. ^1^ Somatic cell count. ^2^ Days in Milk.

**Figure 2 animals-13-00080-f002:**
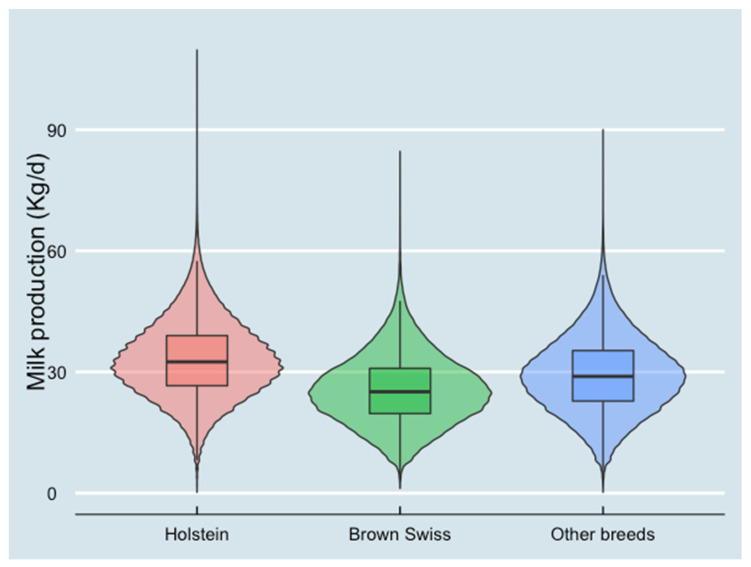
Milk production at test-day records level in different breed groups (10,445,464 test-day records from 826,831 cows in 2970 herds). The violin plot combines the probability density (area under the curve) and box plots.

**Figure 3 animals-13-00080-f003:**
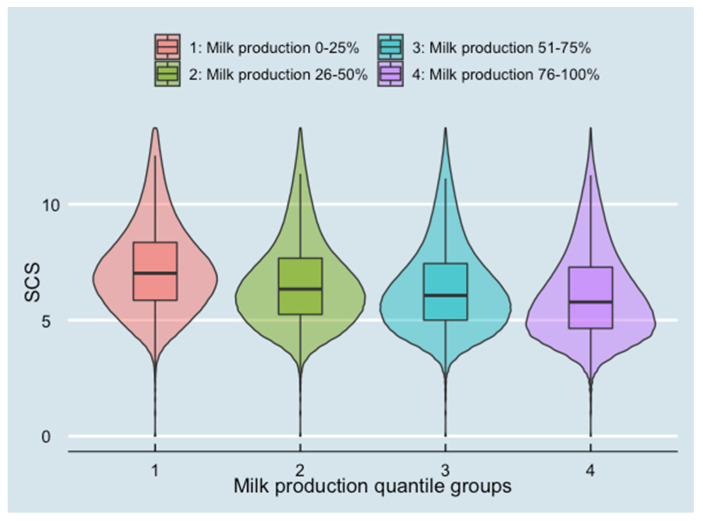
Somatic cell score (SCS) at test-day records level in different milk production quantile groups (10,445,464 test-day records from 826,831 cows on 2970 herds). The violin plot combines the probability density (area under the curve) and box plots.

**Figure 4 animals-13-00080-f004:**
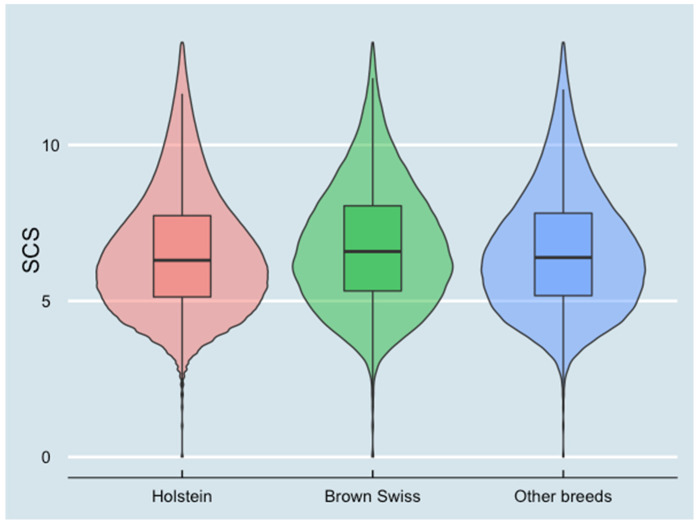
Somatic cell score (SCS) at test-day records level in different breed groups (10,445,464 test-day records from 826,831 cows in 2970 herds). The violin plot combines the probability density (area under the curve) and box plots.

**Figure 5 animals-13-00080-f005:**
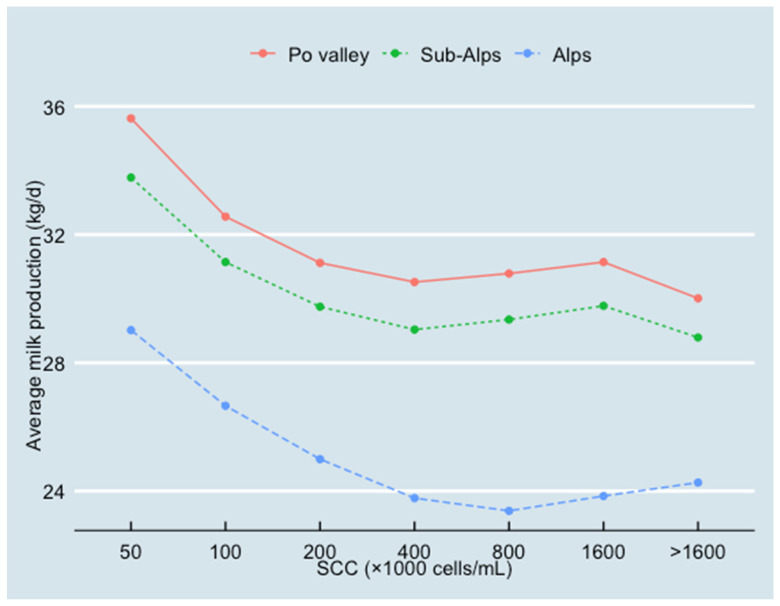
Relationship between SCC and average milk production (10,445,464 test-day records from 826,831 cows in 2970 herds) classified by the three geographical areas (Po Valley, sub-Alps and Alps).

**Figure 6 animals-13-00080-f006:**
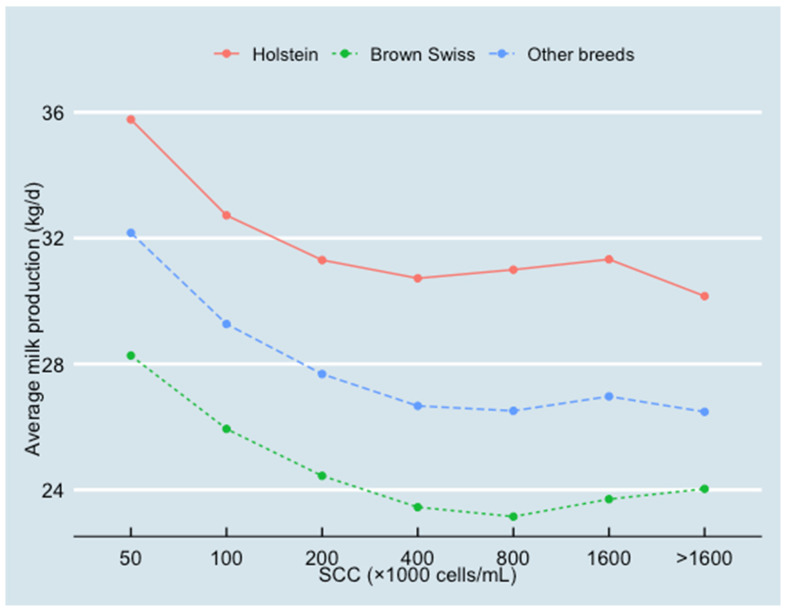
Relationship between SCC and average milk production (10,445,464 test-day records from 826,831 cows in 2970 herds) classified by different breeds (Holstein, Brown Swiss and other breeds).

**Table 1 animals-13-00080-t001:** Distribution of samples among geographical areas and breeds.

Geographical Area	Breed Type	% of Total Test Day Records
**Po Valley**	Holstein	94.2%
	Brown Swiss	1.0%
	Other breeds	4.8%
**Sub-Alps**	Holstein	85.8%
	Brown Swiss	7.1%
	Other breeds	7.2%
**Alps**	Holstein	42.8%
	Brown Swiss	35.2%
	Other breeds	22.0%

**Table 2 animals-13-00080-t002:** Coefficient estimates (β) of the linear mixed model for milk production (in kg/day). The estimates for Days in Milk are not shown.

Independent Variables		Estimated Coefficients (β)	Lower 95% CI	Upper 95% CI
**Intercept**		34.750 ***^2^	34.593	34.907
**SCS ^1^**		−0.830 ***	−0.832	−0.828
**Area**	Po Valley	Ref. ^3^		
	Sub-Alps	−1.354 **	−2.385	−0.324
	Alps	−6.750 ***	−7.386	−6.115
**Breed**	Holstein	Ref.		
	Brown Swiss	−4.639 ***	−4.761	−4.516
	Other breeds	−1.447 ***	−1.500	−1.394
**Parity**	Parity 1	Ref.		
	Parity 2	3.588 ***	3.576	3.599
	Parity 3+	5.112 ***	5.095	5.128
**Season**	Autumn	Ref.		
	Spring	1.912 ***	1.902	1.922
	Summer	0.228 ***	0.217	0.239
	Winter	1.427 ***	1.416	1.437
**Year**	2016	Ref.		
	2017	0.153 ***	0.143	0.164
	2018	0.338 ***	0.325	0.351

^1^ Somatic cell score; ^2.^
*p* < 0.05; ** *p* < 0.01; *** *p* < 0.001; ^3.^ this category is used as reference category in the regression analysis.

**Table 3 animals-13-00080-t003:** Descriptive statistics of the final data (10,445,464 test-day records from 826,831 cows on 2970 herds in Lombardy, including 264,195 test-day records from the Alps, 108,287 test-day records from Sub-Alps and 10,072,982 test-day records from Po valley) at the test-day record level.

	Overall					Alps					Sub-Alps					Po valley				
	Min.	Median	Mean	Max.	SD	Min.	Median	Mean	Max.	SD	Min.	Median	Mean	Max.	SD	Min.	Median	Mean	Max.	SD
**Herd size (nr of cows)**	10	176	228	1099	179	10	47	86	369	93	30	89	109	269	67	30	179	233	1099	179
**Milk production (kg/d)**	0.2	32.2	32.8	109.7	9.6	0.6	25.7	26.3	92.6	9.1	1.2	30.9	31.5	87.6	9.2	0.2	32.4	32.9	110	9.6
**SCC^1^ (x 10^3^ cells/mL)**	1	80	308	9999	779	1	91	319	999	737	1	73	274	9938	710	1	80	308	9999	780
**Fat (%)**	2.0	3.8	3.9	7.0	0.8	2.0	4.1	4.2	7.0	0.8	2.0	3.9	3.9	7.0	0.7	2.0	3.8	3.9	7.0	0.8
**Protein (%)**	1.5	3.4	3.4	6.0	0.4	1.6	3.6	3.6	6.0	0.4	2.0	3.1	3.4	5.8	0.4	1.5	3.4	3.4	6.0	0.4
**DIM^2^ (day)**	5	162	170	399	102	5	158	168	399	104	5	162	170	399	103	5	158	168	399	104

^1^ Somatic Cell Count; ^2^ Days in Milk.

## Data Availability

Restrictions apply to the availability of these data. Data were obtained from Associazione Regionale Allevatori della Lombardia, and are available from the authors with the permission of Associazione Regionale Allevatori della Lombardia (a.ferla@aral.lom.it).

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
