# Peer review of "Linear Mixed-Effects Model to Quantify the Association between Somatic Cell Count and Milk Production in Italian Dairy Herds"

_animals, 2022, doi:10.3390/ani13010080_

Round 1
Reviewer 1 Report
Animals: “Linear mixed-effects model to quantify the association between somatic cell count and milk production in Italian dairy herds” by T. Luo et al.
The authors studied the relationship between somatic cell counts (SCC) and milk production using test day (TD) milk production records. Records were from >800,000 cows in 2,970 herds in 3 geographic regions of Lombard in Italy: The Po Valley, the Sub-Alps, and the Alps. A mixed-effects model with 6 fixed factors (such as breed, days in milk, parity, etc.) was used. The study confirmed the negative association of SCC with milk production. The findings deviated from those reported previously in the literature and provide a clear indication that local factors should be considered when assessing milk loss associated with SCC’s.
This study provides potentially useful information for the dairy industry in many areas in the world and suggests that local factors should be considered in determining milk loss associated with elevations in SCC.
The authors can consider the following comments:
Title, p. 1, l. 2: Minor point, but should “Mixed-effects” be modified to lower case “m”: “Linear mixed-effect model….”?
Simple Summary, p. 1, ll. 14-16: Consider modifying sentence to: “This study confirmed a negative association of somatic cells with milk production based on milk production records……” This puts the main point of the sentence in the first part.
Abstract, p. 1, l. 33: There is probably not another good way to do it, but the presentation of milk production loss here is confusing until you think through it: “….the milk production lost 0.830 (95% CI: -0.832, -0.828) kg/cow/d…”
Introduction, p. 1, l.41: Minor point, but this sentence sounds better as “It is known that these costs can vary (delete “a lot”) considerably between farms [5-7].”
Introduction, p. 1, l. 42: Replace “can achieve” with “can reach”?
Introduction, p. 2, l. 45: Should “contagious-free herds” be “contagious pathogen-free herds.”?
Introduction, p. 2, l. 57, l. 59, and probably other places: Minor point, but: In “..4.70kg for SCC..” and “..919kg per lactation…” there is no space between the number and the kg.
Introduction, p. 2, l. 59: Juliano et al. [15] reference should be Goncalves et al. [15].
Introduction, p. 2, l. 69: Minor comment & just a thought: Would a map showing where these regions are located be interesting to the reader?
Introduction, p. 2, l. 69: Minor point again, but modify “…large scales intensive dairy herds..” by removing s from “scales” and making it “scale.”
M&M, p. 2-3, ll. 94-96: Just for clarification, how did you decide on the specifics of excluding herds with <30 cows for 2 of the regions and <10 cows for the Alps? It would be good to give the reader a short explanation.
M&M, p. 3, Figure 1: Please check the numbers for “records”: I tried multiple times to follow number of records from the beginning to the end—and I could not make it work out to the final numbers you show. I might be missing something—but it doesn’t look right to me.
M&M, p. 4, Minor point: Equation between l. 132 and 134: “Heard” is misspelled—should be “herd,” as you know.
Results, p. 4, l. 166: Minor point again, but “..103 cells/ml…” should be 103 cells/mL..”
Results, p. 4-5, Table 1: Again, a minor point, but Table 1 is hard to read—hopefully it will set out in a more readable manner when published.
Results, p. 8, l. 204: Minor comment again—but same comment as for Abstract, p. 1, l. 33.
Results, p. 8, l. 208: The meaning of this sentence is not clear to me: “In each year, considered in this study, milk production increased gradually.” Taken literally, this could be interpreted as milk production increasing within herds during a given year. Or, are you referring to milk production increasing from year to year (e.g., from 2016 to 2017), within the period of the study, like from 2016 to 2018.
Results, p. 8, l. 211: First word on line should be “indicating” as opposed to “indicated.”
Results, p. 8, l. 219: “..the milk production was smaller 35 kg/day…” Presumably this should state that “the milk production was less than 35 kg/day…”
Discussion, p. 9, l. 237: Misspelling of “averge”—should be “average.”
Discussion, p. 10, l. 310: First whole sentence: “These information should be considered….” I believe that this should be “This information….”
Conclusions, p. 10, l.321: “milk production lost 0.830 (95% CI: -0.832, -0.828)..” Same comment as for Abstract, p. 1, l. 33.
Author Response
Please read the cover letter for all the answers to Reviewer's comments.

Reviewer 2 Report
The manuscript under review explored the association between somatic cell count and milk production in Italian dairy herds. This is a well-researched area. The available scientific literature has established the fact that increasing somatic cell count deceases milk yield. The current study has a very little information that would add in the existing knowledge. However, the findings of the current manuscript could be important for the local dairy industry in Italy and other areas with similar dairy production systems. Some suggestions are listed below to improve the manuscript.
Introduction
The introduction section was well constructed to establish the need of the study. Minor edits are suggested as follows.
Line 62: keep this with the previous paragraph for the flow of the story.
Line 70: Separate the objectives as a new paragraph. Add a hypothesis if possible.
Line 73-76: These are the implications of the study and should come after the conclusion paragraph.
Materials and Methods
Line 88-91: This information should come under a subheading “Study data” and be the first subheading in materials and methods section. Add all the information about the data here, then describe the standard sampling methods of the association.
Data editing was well explained.
Line 101-115: The records percentage should be descriptively presented in results section in a table form as follows.
Table. Distribution of test day records by breed, area, parity, season, and herd size (n= 10.44 million)
|
Variable |
% Of test day records |
|
Breed |
|
|
Holstein |
|
|
Others |
|
|
Area |
|
|
A |
|
|
B |
|
|
Parity |
|
|
1 |
|
|
2 |
|
… and so on.
Line 122-124: Please clarify this sentence. Use simple narrative like SCC was converted to linear somatic cell score using base 2 log transformation of SCC.
Explanatory variables.
It is strongly recommended to add the herd size as a fixed effect in the model. Determine the rolling herd size. Make rationale categories of herd size and use it in the model.
Line 167: Rearrange the table as follows.
|
Variable |
Means ± SD |
Range |
|
Herd size, n |
|
|
|
Milk yield, kg |
|
|
|
SCS or SCC, |
|
|
Line 176: Please describe the figure properly; what do the box, the colored areas, extended lines, and the width of the colored area tell? Otherwise present the information in tables. Same comment for Figure 3 and 4.
Lines 192 and 196: Present and describe the figures 5 and 6 after the Table 2.
Line 198: While determining the association between the variables in an observational study, one needs to use the mixed models to remove the bias and confounding. Therefore, the subheading should be to describe the association between SCS and milk yield, area, parity, and breed.
Line 209-223: This information is just the details of the statistical analyses and may be presented in the supplementary file.
Line 231-241: This explains that herd size in different areas was different. This further strengthens the fact that the difference in SCC and milk yield could be due the herd size variation instead of the area. Therefore, it is necessary to add the herd size in the model to determine the actual association between the SCS and the milk yield and/or the area.
Line 280-292: The discussion related to model fit may be presented in the supplementary file.
Conclusion
Please focus on the main findings in the conclusion section. To include information about methodology in conclusion is less appropriate.
Author Response

(The authors gave the same response as above.)

Round 2
Reviewer 2 Report
The authors have revised the manuscript and have acceptably addressed the concerns identified in the initial review.